# MiRNAs in Canine and Human Osteosarcoma: A Highlight Review on Comparative Biomolecular Aspects

**DOI:** 10.3390/cells10020428

**Published:** 2021-02-18

**Authors:** Leonardo Leonardi, Katia Scotlandi, Ilaria Pettinari, Maria Serena Benassi, Ilaria Porcellato, Laura Pazzaglia

**Affiliations:** 1Reparto di Patologia Generale e Anatomia Patologica Veterinaria, Dipartimento di Medicina Veterinaria, Università degli Studi di Perugia, 06126 Perugia, Italy; pettinariilaria@hotmail.it (I.P.); ilariaporcellatodvm@gmail.com (I.P.); 2Laboratory of experimental Oncology, IRCCS—Istituto Ortopedico Rizzoli, 40136 Bologna, Italy; katia.scotlandi@ior.it (K.S.); mariaserena.benassi@gmail.com (M.S.B.); laura.pazzaglia@ior.it (L.P.)

**Keywords:** osteosarcoma, comparative oncology, dogs, humans, biomolecular, miRNAs

## Abstract

Osteosarcoma (OS) is the most frequent primary malignant tumor of bone in humans and animals. Comparative oncology is a field of study that examines the cancer risk and tumor progression across the species. The canine model is ideally suited for translational cancer research. The biological and clinical characteristics of human and canine OS are common to hypothesize as that several living and environmental common conditions shared between the two species can influence some etiopathogenetic mechanisms, for which the canine species represents an important model of comparison with the human species. In the canine and human species, osteosarcoma is the tumor of bone with the highest frequency, with a value of about 80–85% (in respect to all other bone tumors), a high degree of invasiveness, and a high rate of metastasis and malignancy. Humans and dogs have many genetic and biomolecular similarities such as alterations in the expression of p53 and in some types of microRNAs that our working group has already described previously in several separate works. In this paper, we report and collect new comparative biomolecular features of osteosarcoma in dogs and humans, which may represent an innovative update on the biomolecular profile of this tumor.

## 1. Introduction

MicroRNAs (miRNAs) are highly conserved non-coding RNAs of ~22 oligonucleotides that exert a primary modulatory role on gene expression and can regulate more than 60% of the genome post-transcriptionally. They have been characterized in 61 tissues [1] and their presence has been discovered in 12 human body fluids [2]. The first descriptions of the roles played by microRNAs date back to the 1980s when they were identified by Chalfie in 1981, and, over the years, the numerous studies applied to oncology have shown how their deregulation can play a significant role in the etiopathogenetic mechanisms of some of the most common forms of cancers, both in humans and animals. The comparison of the expression of microRNAs among neoplastic and normal cells has further established their role in the oncogenesis of many tumors by means of interaction and promotion mechanisms on target genes [3]. Moreover, the miRNAs alterations often correlate with a more aggressive tumor biological behavior, making these molecules potential prognostic and therapeutic targets [4].

Comparative oncology is a field of study that assesses the cancer risk and tumor progression across species. The canine model is ideally suited for translational cancer research. Some histological and clinical characteristics of numerous canine tumors are strongly similar to those of the corresponding tumors in humans. The common dog’s lifestyle practices, in an environment totally shared with the human species and where the development of spontaneous tumors can be just as common, places the dog in a unique position that can allow for a better understanding of the development and progression of cancer than traditional models of laboratory animals [5].

Canine sarcomas represent an excellent comparative model to those of the human species for a series of reasons, including biological similarities between the two species that arise spontaneously, and also that they have in common many external risk factors and lifestyles.

## 2. Osteosarcoma

Osteosarcoma (OS) is a high-grade malignant tumor of bone composed of mesenchymal cells (malignant osteoblasts) that is able to produce an immature woven bone and osteoid matrix. Primary osteosarcoma of a bone arises in the mesenchymal tissue that forms the cortical bone or is present in the medullary cavity of any bones. Conventional OS is divided into different subtypes according to the characteristics of the tumor and the predominant histological differentiation (osteoblastic, fibroblastic, chondroblastic, small cell, high-grade telangiectatic, and extra skeletal surface).

Some of the comparative age-related biological characteristics of OS between humans and animals, especially in dogs, are interesting. In humans, two peaks of occurrence are described. The largest peak occurs in young patients between 10 and 20 years old. The second group consists of osteosarcoma arising in Northern Europeans secondarily to predisposing diseases such as Paget’s disease of bone, or Li-Fraumeni syndrome. In dogs, the age range for diagnosis of osteosarcoma in epidemiological studies is reported to be from 3 months to very old large-breed dogs, with most of the diagnoses regarding adult animals. In humans, osteosarcoma occurs much more frequently in very young and adolescent subjects, with variable secondary frequency peaks in adults over 65 years of age. In humans, the tumor is mainly localized to the long bones such as the distal femur, the proximal tibia, and the humerus [6]. In the canine species, the different forms of OS predominantly affect large-breed dogs, with localization of the tumor in the appendicular skeleton. The etiology of this tumor is still unknown, although several etiopathogenetic hypotheses have been formulated, including the deletion of TP53 and Rb, which can cause osteosarcomatous transformation of osteoblasts [7]. No comparable predisposing genetic factors have currently been recognized as tumor-promoting factors in dogs or other domestic animals. In veterinary medicine, osteosarcomas are essentially classified in three categories on the basis of the primary localization of the tumor: central, parosteal, and periosteal, where diagnostic features are characterized by production of osteoid and tumor bone and embedded malignant osteogenic cells. In canine and feline osteosarcomas, central osteosarcomas have the highest degree of malignancy with a frequent and fast metastasizing process to the lungs. The genetic susceptibility in human OS is associated with heritable cancer syndromes. The most frequent cases of diseased or predisposed bone are represented in veterinary medicine by trauma, orthopedic implants (foreign bodies), specific genetic abnormalities, chemotherapy, and radiotherapy. There are still very few data available on the etiopathogenesis and the biomolecular and genetic characteristics, such as predisposing and causal factors determining canine osteosarcoma.

The frequency of chromotripsis (a mutational process by which up to thousands of chromosomal rearrangements occur grouped into a single event in localized genomic regions confined to one or a few chromosomes) in OS is high, with a frequency value of approximately 77% [8]. This anomaly in OS generates amplification of genes such as CDK4, MDM2, COPS3, RICTOR, and TERT gains, or disruption of driver oncogenes (TP53, NF) [8].

Treatment of OS in both humans and dogs requires a multidisciplinary approach involving the combined action of surgery with preoperative and postoperative chemotherapy using cytotoxic factors (cisplatin, doxorubicin, high-dose methotrexate/ifosfamide) [9]. Therapeutic advances with neoadjuvant and adjuvant chemotherapy have led to improved overall survival rates in OS patients with non-metastatic disease at diagnosis by up to 70%, but for metastatic patients at diagnosis and with relapse survival rates, they are only 20% [9].

This working group has been working on biomolecular investigations to try to identify in a comparative way some of these intimate aspects characterizing the pathogenesis of osteosarcomatous diseases in humans and dogs. We have already demonstrated some human and canine OS similarities both in original studies and in unpublished investigations. In this review, we will summarize and focus on the analyses of the variations in expression of some miRNAs we tested and that are still under study.

## 3. MiRNA in Canine and Human Osteosarcoma: Comparative Features

Several studies have investigated the role of microRNAs in human OS by means of miRNA expression profiles. Altered miRNA expression and a unique miRNA signature have been identified and associated with the risk of metastasis and a specific response to chemotherapy [10,11,12,13,14]. Since the discovery of miRNAs, few studies have dealt with the association between deregulation of miRNAs and canine OS [15]. Nevertheless, these studies and our own previous studies evaluating miRNA deregulation in naturally occurring canine cancer (such as spontaneous OS), demonstrate that, similarly to its human counterpart, aberrant miRNA expression likely contributes to tumor biology and progression [16,17,18,19].

Thayanithy et al. postulate that multiple microRNAs present at the 14q32 locus in human OS, compared with normal bone tissue, result in downregulation, and that the epigenetics modifications of this locus may contribute to the related alterations. The combinatorial treatment with DNA and chromatin-modifying drugs (such as 5-AzadC) may also activate different miRNAs at the 14q32 locus and significantly modify the lower expression of cell-cycle genes in treated Saos2 OS cell cultures [20]. Sarver A. et al. confirm—on a series of human OS tissue and canine OS tissue and cell lines— 14q32 miRNA downregulation, using miR-382 as a representative of 14q32 miRNAs in human OS and miR-134 and miR-544 in canine OS. In particular, they show for both species an evident association between 14q32 miRNAs decreased expression level and poor outcome in OS patients. Those data suggest that the dysregulation of the 14q32 miRNA cluster may represent a conserved mechanism responsible also for the aggressive and invasive biological behavior of OS in both humans and dogs [21].

MiR-196 is located in the regions of homeobox (HOX) clusters and could be involved in the regulation of those genes (HOXC8, HOXB8, HOXD8, and HOXA7) by playing an important role in cellular development [22]. Its overexpression was assessed in several human cancers, in particular glioblastomas [23], hepatocellular carcinoma [24], and colon cancer [25]. Yang et al. show a downregulation of miR-196 in OS cell lines and postulate that this miR inhibits HOXA9 to promote proliferation and migration of human OS cells [26]. Pazzaglia et al. confirm a downregulation of miR-196a in OS of both species, describing an increase of its target, Annexin 1, in tissues and cell lines. The effects of miR-196a overexpression on tumor cell response may be strictly related to species and cell type. The ectopic expression of miRNA-196a in cell lines seems to influence the significant decrease in cell proliferation and also the increase in apoptotic phenomena, especially in the human cell line 143B of OS. From our study, we identified a transient decrease in some cell motility factors in the human OS 143B cell line and canine OS cell line (DAN), and a more sustained decrease in the other human OS cell line (MG63) [27].

Fenger JM et al.in a series of canine OS cell lines and tissues, identified the miRNA expression using the nanoString system of analysis observing 26 overexpressed miRNAs in canine OS tumor samples compared with normal canine osteoblast cells, and about 44 others miRNAs that insetad were downregulated.

MiR-1, miR-9, miR-10b, miR-29a, miR-122, miR-126, miR-199b, miR-200c, and miR-451, all mature miRNAs that share 100% sequence homology between dogs and humans, were afterwards validated. These miR-9 expression levels were found to be significantly higher in primary canine tumor samples when compared with normal canine osteoblasts [28]. Different studies assessed that miR-9 is downregulated in several human and canine cancers (esophageal, ovarian, colon, renal, gastric, etc.) [29,30,31,32], whereas other investigators observed an overexpression of this miRNA in other forms of cancer such as biliary tract cancer, breast cancer, brain tumor, and lung cancer [33,34,35,36].

Gao et al. show that inhibition of miR-9 decreases the OS cell proliferation by targeting p16; meanwhile, Zhu et al. evidence this role as oncogene-promoting proliferation by targeting Grap2 and cyclin D interacting protein [37,38]

Over the last several years, the role of miR-9-5p in breast cancer, osteosarcoma, and hepatocellular carcinoma has been evaluated, and its overexpression of miR-9-5p has shown correlations with advanced tumor stages and poor prognosis [39,40]. A study about a large series of OS specimens and corresponding non-cancerous bone tissues revealed a miR-9 expression increase, and this high expression is associated with the worst outcome [41].

Our group has previously shown that the overexpression of miR-9 in canine OS is not secondary to non-neoplastic cells infiltrating the tumor microenvironment, but was indeed produced by malignant osteoblasts. The importance of miR-9 in tumors seems to be conformed also in canine mast cell tumors, where Fenger’s group has shown that miR-9 overexpression is associated with an aggressive, metastatic behavior in primary canine mast cell tumors. These data suggest that the expression of miR-9 in malignant mast cells may enhance their invasive capacity by inducing the expression of genes that promote cellular invasion [28]. However, these data about canine OS indicate that, while enforced miR-9 expression in normal canine osteoblasts and the OSA16 cell line enhanced cellular invasion and migration, miR-9 had no impact on cellular proliferation or viability.

In our previous retrospective study on several human and canine OS specimens, we demonstrate low levels of miR-1 and miR-133 b in OS samples when compared with normal bone. These findings were also associated with positive immunostaining for MET protein, characterized by strong nuclear expression in more than 50% of OS cells and with the protein expression of MCL1 in 72% of examined tumors. These results, in association with the higher expression of the target MET and MCL1 genes, suggest a similar pathway involvement in osteosarcoma development in both human and canine species [42].

Among the various miRNAs considered in the etiopathogenetic and biological mechanisms of progression of many tumors, the miR-34 family (miR34a, miR-34b, and miR-34c) has been investigated for a long time, obtaining important results that suggest that members of this family play an important role as tumor suppressors in a wide variety of human spontaneous tumors [43]. It has now been established that these miRNAs can interfere with the mRNA of various cellular proliferative and anti-apoptotic regulatory factors by negatively controlling their expression, which can thus lead to cell cycle arrest, cell senescence, and apoptosis [44]. It is now also clear that p53 trans activates the microRNAs (miRNAs) of the 34 families. Several studies have shown a decrease in miR-34 expression in human OS associated with an increase in several target genes known to be involved in the mechanisms of tumorigenesis, such as MET, SIRT1, and CDK6 [45,46]. Lopez et al. demonstrated that miR-34a expression is significantly reduced in primary canine OSA tissues and cell lines compared with normal canine osteoblasts. Stable overexpression of miR-34a in canine OS cell lines also reduced the expression of angiogenic factors such as VEGFA (Vascular Endothelial Growth Factor) with a concomitant consequent decrease in cell invasion and migration [47]. Furthermore, lower expression of mi-34a in tumor and plasma has also been associated with poor prognosis and chemo resistance, and, recently, a new bioengineered tRNA/miR-34a prodrug demonstrated important antitumor activity in a canine model of OS [4].

Some of our previous data show increased expression of miR-106b and miR.93-5p in human OS, although OS p53wt cells responded to ectopic overexpression of miR-93-5p more significantly, resulting in increased proliferation and tissue invasion compared with cells lacking functional p53. Analysis of the miR-106b cluster (miR-106b, miR-93-5p, and miR-25) in human and canine OS shows variable expression of these molecules. No significant difference with corresponding normal bone was observed in miR-106b and miR-25 expression, while miR-93-5p expression was increased in all OS samples, with higher expression levels in the canine subgroup compared with the human subgroup [48,49] (Figure 1). Zhang et al. show that miR-93-5p, via p21, increases cell proliferation in some forms of nasopharyngeal carcinoma, while inhibiting apoptosis in human hepatocellular carcinoma [50]. In our study, after the downregulation of miR-93-5p in both subgroups, p21 expression was increased at both the mRNA and protein levels and, in particular, the introduction of the inhibitor miR-93-5p caused a cellular response in cultures of human OS 143B and canine OS DAN, which differed in the most intense functional impact in DAN.

## 4. Circulating MiRNAs

Accumulating evidence indicates that circulating tumor non-coding RNAs are also promising biomarkers for early diagnosis and prognosis in cancer. The miRNAs that are secreted or that leak from cancer cells are either present in vesicles formed by the lipid bilayer (extracellular vesicles) or form a complex with proteins and lipids and remain present in the blood without being degraded by RNase.

In human OS, several circulating miRNA global profiling studies were performed, and a deregulation in many of them was identified. The presence of OS was in fact associated with increased levels of circulating miRNAs, including 14q32 miRNAs, miR-95-3p, miR-300, miR-101, miR-542-3p, miR-196a, miR-196b, and miR-491, and with decreased levels of miR-17, miR-497, miR-106a-5p, miR-16-5p, miR-20a-5p, miR-425-5p, miR-451a, miR-25-3p, and miR-139-5p [51].

Several reports show an association between miR-214 and miR-126 with the pathogenesis of human OS by increasing the tumor cell’s proliferation, survival, metastasis, and chemo resistance [52]. An increase in these molecules was also seen in human and canine OS [53]. In particular, the authors demonstrate higher levels of circulating miR-214 in dogs affected by OS when compared with other dogs affected by non-epithelial tumors. Additionally, an in vivo study suggests an association between miR-214 and tumorigenesis in OS. Moreover, the same group showed that levels of circulating miR-214 and miR-126 before treatment could predict time to metastasis after amputation and chemotherapy in dogs with appendicular OS [54]

## 5. Discussion and Conclusions

MiRNAs are highly conserved non-coding RNAs of ~22–25 oligonucleotides that exert a primary modulatory role on gene expression, regulating more than 60% of the genome post-transcriptionally. Over the years, biomolecular studies on the expression of some biomarkers, such as microRNAs, have allowed us to identify forms of altered regulation of the same type in different forms of spontaneous tumors; in these studies, it has been established how these can play the role of tumor suppressor or tumor promoter in the co-involvement of specific targeting genes, resistance to apoptotic phenomena of cell death, the control of angiogenesis, etc. The correlation between these altered expressions of microRNAs has also allowed us to understand how these can play a fundamental role in the identification and application of new therapeutic models.

Moreover, miRNA alterations correlate with a more aggressive behavior of cancer, making these molecules potential prognostic and therapeutic targets.

In the last 20 years, comparative oncology has quickly expanded and the study of spontaneous cancers in the domestic dog has provided a suitable and interesting model for understanding several etiopathological mechanisms and, above all, new diagnostic tools and therapeutic paths for cancer in humans.

We are confident that continuing on this template in the exploration and study of new markers will allow us to identify innovative and hitherto unknown aspects that will allow us to establish new genetic and biomolecular investigation methods for an increasingly early diagnosis of osteosarcoma, as well as to establish new methodologies for an increasingly targeted and decisive therapeutic approach. We are also firmly convinced that the comparative study of these parameters allows for an approach to these issues with a broader view and consideration of fundamental comparative factors between species, which recognizes the canine study model as an important reference factor for also understanding the oncological and osteosarcomatous dynamics of the human species. We know well how the new therapeutical approaches respect conventional treatments, and are clearly required, especially when involving biomolecular and personal pharmacogenomics therapies, using specific biomarkers and biomolecular targets, and the use of animals for comparative therapeutic considerations; these approaches can also represent a valid system of comparison and reference for the understanding of the etiopathogenetic mechanisms and progression paths of deep cell tumor transformation. Further studies are required to establish and improve treatments in both humans and dogs. Additionally, it is still necessary to investigate specific genetic modifications in canine and human OS to try to detect new genetic loci modifications. Anyway, we would like to underline the high potential of miRNAs as prognostic biomarkers for canine and human osteosarcoma; therefore, further studies comparing overall survival and disease-free interval after treatment are necessary. Circulating miRNAs have recently been suggested to be an important diagnostic and prognostic biomarker in several human cancers, including OS. The different variations on increased levels of different miRNAs permit us to also identify in dogs the high diagnostic accuracies and partially detect in different types of sarcomas and epithelial tumors. Our studies confirm and demonstrate an increase in the levels of miRNAs investigated, suggesting a good diagnostic accuracy that was not well detected in comparative studies before. Future studies will be necessary to develop new strategies to improve the diagnostic and prognostic potential of these miRNAs in canine and human OS.

Our working group already boasts long-standing collaborations that have allowed us to obtain interesting results on the comparative expression of different isoforms of miRNAs in human and canine OS, such as the miR-196a, miR-1, miR-133b, and miR-106B-25 cluster, confirming the biomolecular similitude between canine and human OS and suggesting a potential role of miRNAs as new biomarkers for OS early diagnosis and treatment.

From what can be extrapolated from the data in the literature, there is still no further information relating to the expression of factors altering the activity of miR-9 and other miRNAs, and this is all we have been able to obtain up to now from our investigations and recognitions in the related data banks. This lack of data, however, will quickly represent a further stimulus to continue on the path of comparative biomolecular evaluation on these and other factors, which to date still remain unexplored, as well as those of several exosomes, premetastatic niches, and other factors that we have set out to investigate together in future comparative works on human and canine osteosarcoma. Future studies of these factors and those of other biomolecular and genetic aspects—aimed at an increasingly sophisticated characterization of osteosarcomas—are still so devastating and lethal for many of their biological aspects and will certainly guarantee to be able to identify new innovative diagnostic and therapeutic paths, in order to guarantee the identification of new access “keys” for ever more early and precise diagnoses and for ever more targeted and decisive therapies. The animal model, also from a “One Health” perspective, will be able to ensure for a long time a study model on which to build new comparative safeguards for the fight against many diseases, including those of a more difficult approach, such as those of oncological type.

## Figures and Tables

**Figure 1 cells-10-00428-f001:**
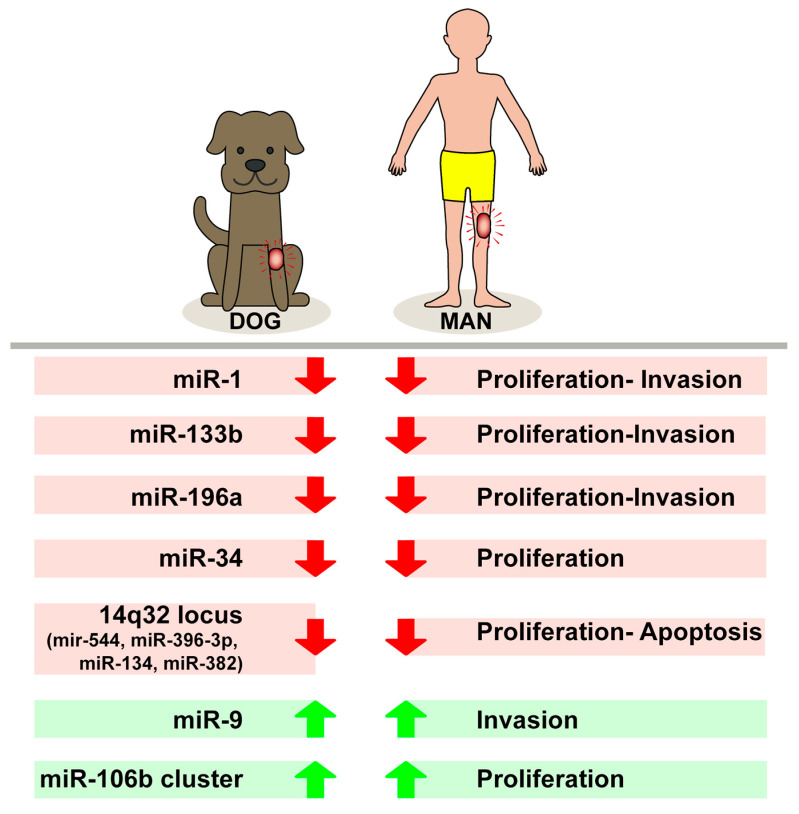
Different levels of expression of tested microRNAs in biological behavior of canine and human osteosarcomas.

## Data Availability

Not applicable.

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
