# Peer review of "MiRNAs in Canine and Human Osteosarcoma: A Highlight Review on Comparative Biomolecular Aspects"

_cells, 2021, doi:10.3390/cells10020428_

Round 1
Reviewer 1 Report
See the attachment

Author Response
Dear Reviewer, we would like to thank you very much for your precious suggestions. We have tried to correct and improve all the aspects suggested by you, in the hope of having correctly answered what was requested.

Reviewer 2 Report
Manuscript Cells-1111875
Reviewer comments
The manuscript entitled "Canine and Human Osteosarcoma: a highlight review on some of new comparative biomolecular aspects" by Leonardo Leonardi et al., is an interesting review comparing miRNAs aspects between human and dog osteosarcomas. The review covers the various implications of miRNAs in osteosarcoma tumor genesis with a systematic comparative approach between man and dog. This review is well structured considering successively tumor cell expressing miRNAs and circulating miRNAs.
The reviewer has no comment on this review. Some may wonder why the term miRNAs, corresponding to the central subject of the review, is not present in the title.
Whatever, the Reviewer recommends to accept this manuscript after an edition of the text for English usage.
Author Response

(The authors gave the same response as above.)

Reviewer 3 Report
In this paper, the authors reviewed the correlation of miRNA expression abnormalities in human and canine osteosarcoma. This review is very informative and the conclusion is adequate, however, following comments on Title and Figure 1 need to be improved by the authors:
1) Since this review focuses on only miRNAs, it is better to include miRNAs in the title.
2) In Figure1, it is recommended that the upregulated miRNA is shown in the upper row and the downregulated miRNA is shown in the lower row.
Author Response

(The authors gave the same response as above.)
